# Influence of Hazelnut and Walnut Oil Cakes Powder on Thermal and Rheological Properties of Wheat Flour

**DOI:** 10.3390/foods12224060

**Published:** 2023-11-08

**Authors:** Karolina Pycia, Lesław Juszczak

**Affiliations:** 1Department of Food Technology and Human Nutrition, Institute of Food Technology, College of Natural Science, University of Rzeszow, Zelwerowicza Street 4, 35-601 Rzeszow, Poland; 2Department of Food Analysis and Evaluation of Food Quality, University of Agriculture in Krakow, Balicka Street 122, 30-149 Krakow, Poland; rrjuszcz@cyf-kr.edu.pl

**Keywords:** wheat flour, hazelnut oil cake, walnut oil cake, DSC differential scanning calorimetry, rheological properties

## Abstract

The aim of the study was to assess the influence of the addition of powdery hazelnut oil cakes (HOC) or walnut oil cakes (WOC) to wheat flour (WF) on its selected thermal and rheological properties. In the research material, part of the wheat flour (5%, 10%, 15%) was substituted with powdery oil cakes based on hazelnuts and walnuts. The control sample was wheat flour (100% WF). In the tested systems with the addition of hazelnut oil cakes (WFHOC) and walnuts (WFWOC), the characteristics of the gelatinization and retrogradation processes were determined using the DSC method, the gelatinization characteristics of 10% pastes using the RVA method, flow curves and viscosity curves, as well as mechanical spectra. Based on the results obtained, it was found that the type of oil cakes and the level of their addition significantly influenced the thermal and rheological properties of the tested systems. Partial replacement of wheat flour with HOC or WOC significantly influenced most DSC parameters. The highest values of gelatinization enthalpy ∆H_G_ and retrogradation ∆H_R_ were characteristic of the WFWOC5% sample (5.9 J/g) and the control sample (1.3 J/g), respectively. All tested systems showed the properties of shear-thinning non-Newtonian fluids, and the partial replacement of wheat flour with HOC or WOC resulted in a significant reduction in the maximum viscosity of pastes, increasing with the increase in the proportion of oil cakes. WFHOC-based pastes were characterized by higher values of the G′ and G″ modulus, while their values and the values of the K′ and K″ parameters decreased as the share of oil cakes increased. Gels based on all tested systems showed the nature of weak gels (tan δ = G″/G′ > 0.1). Replacing part of the wheat flour with nut oil cakes modified the thermal and rheological properties of pastes and gels, and the observed changes were influenced by both the origin and the level of addition of powdered oil cakes. It was found that WFHOC/WFWOC15% systems had reduced viscosity and weakened viscoelastic properties compared to systems with a lower OC content, which is not a favorable feature from the technological point of view. However, these systems were the most stable, which is an advantageous feature. However, for baking purposes, research should be carried out on the rheological properties of dough made from these mixtures.

## 1. Introduction

The growing nutritional awareness of modern consumers makes them look for natural, low-processed food that most closely resembles plant or animal raw materials. Plants are a natural, very rich source of substances with health-promoting potential. Examples of raw materials belonging to this group are, for example, oil seeds such as sunflower, linseed, pumpkin, rapeseed or nuts of various botanical origins. The use of oilseeds is gradually increasing due to consumers’ growing interest in a balanced diet and a healthy lifestyle. Consumers are aware of the relationship between the quality of their diet and the prevention of diet-related diseases [1,2]. The common feature of oil seeds is a very high fat content (>40%) and a number of various biologically active compounds. In turn, compared to cereal grains, they have a lower carbohydrate content but a higher level of protein, fiber, polyunsaturated fatty acids from the omega-3 and omega-6 groups, and other phytochemicals [1]. In terms of chemical composition and health benefits, nuts are interesting, especially hazelnuts and walnuts. Hazelnuts dominate among nuts in terms of the content of α-tocopherols [3]. The favorable chemical composition of walnuts is due to the high content of fatty acids and tocopherols. It has been shown that thanks to the high PUFA content, walnuts contribute to reducing the risk of heart disease by lowering the level of LDL cholesterol and increasing the level of HDL cholesterol [4]. Walnut oil is very popular in the food industry due to its favorable profile of fatty acids, which are essential in the prevention of diseases of the nervous system [5]. Consumers are aware of the health-promoting potential of this category of plant raw materials and include them in their diet. Similarly, food producers compose food with the addition of oilseeds, but in recent years, they have been increasingly using by-products from food production. Pomace, bran, and spent grain are also widely used in the composition of functional foods. The modern trend of using by-products from the food industry (bran, pomace, spent grain, oilseed cake) in food production makes it possible to create a closed-loop economy in which the by-products are reused to create innovative food products. This type of approach is consistent with the assumptions of the European Green Deal, which assumes minimizing the impact of industry on the environment and the need to manage by-products [6]. Nowadays, there is a significant trend related to the valorization or fortification of food products through the use of raw materials or by-products from the food industry rich in bioactive substances for their production. Unconventional raw materials related to the production of walnuts, such as walnut flowers, are also useful for composing new products with antioxidant potential. Pycia et al. [7] developed wheat bread with dried walnut flowers. According to the cited authors, the bread had a high antioxidant potential, which increased with the increased share of the nut ingredient in the recipe. Moreover, other studies have shown that both walnut flowers [8], pollen [9], and leaves [10] are valuable plant raw materials in terms of polyphenol profile, which can be used to compose food products that are innovative in terms of their health-promoting potential. However, the most valuable ingredient of all nuts is fat. Fat is isolated from oilseeds by extraction or pressing. The cold pressing method is more advantageous due to the preservation of temperature-sensitive bioactive substances. A by-product of the process of pressing oil from oilseeds is oil cakes (OC), also known as pomace. Like oilseeds, they can serve as functional ingredients containing oleochemicals, phytochemicals, fiber, minerals and many other health-important ingredients with strong antioxidant properties. Until recently, they were used as animal feed. However, they are now increasingly used in the food, cosmetics, and pharmaceutical industries [11]. Oil cakes are a by-product produced during the extraction or pressing of oil from oilseeds. Oil cakes can be divided into edible and inedible cakes. Edible oil cakes include those based on sunflower seeds, nuts, or linseed. In turn, the by-product from obtaining castor or sesame oil is considered inedible due to the high content of toxic ingredients [12]. According to the cited authors, edible oil cakes are an excellent, rich source of protein, fiber, minerals, vitamins, and other bioactive substances [12], which is why they are often used to improve the nutritional value of baked goods, like bread and others [12]. When added to flour, they will certainly affect its rheological properties related to the gelatinization process of starch, which is its main ingredient. Pycia and Juszczak [13,14] found a significant impact of ground hazelnuts and walnuts on the rheological properties of flour and dough. The nature and direction of these changes depended on their type and the size of the addition. However, it is not known how the addition of the product after pressing nut oil will affect the rheological characteristics of the flour. Assessment of the rheological properties of flour or its mixtures with plant raw materials such as nut oil cakes is of key importance for the bakery industry. It helps predict the characteristics of the dough, its behavior during kneading and the quality of the product after baking. The water absorption of flour, the consistency and efficiency of the dough, as well as the degree of its softening and susceptibility to deformation, will probably change. However, this topic requires further research because the direction of these changes is unknown. 

Therefore, the aim of the study was to evaluate the impact of the substitution part of wheat flour (5%, 10%, and 15%) with hazelnut oil cakes or walnut oil cakes on the selected thermal and rheological properties of this system. The control sample was wheat flour.

## 2. Materials and Methods

### 2.1. Materials

The systems in which wheat flour (WF) (type 650, Złote Pola, Poland) was replaced with powdery hazelnut oil cake (HOC) or walnut oil cake (WOC) were in the amount of 5%, 10%, and 15%. Therefore, the WF content in these systems was 95%, 90%, and 85%. Hazelnut and walnut oil cakes (Warmińska Manufaktura, Jeziorany, Poland) were obtained by cold pressing the fat at a temperature up to 38 °C. HOC and WOC were thoroughly mixed with wheat flour, and the resulting systems (WFHOC5%, WFHOC10%, WFHOC15%, WFWOC5%, WFWOC10%, WFWOC15%) were sifted through a hand kitchen sieve. Homogeneous systems were stored at 5 °C until analysis. The control sample was WF (control) without the HOC and WOC. 

### 2.2. Methods

#### 2.2.1. Determination of Chemical Composition of Analyzed Systems

In the analyzed systems, moisture [15], fat content [16], total ash content [16], and crude fiber content [17] were determined. The moisture content of the samples was tested by drying in a dryer (Binder, Merazet, Poznań, Poland) to constant weight at 130 °C. The fat content was determined using the Soxhlet method (Soxtec 2050, FOSS, Warszawa, Poland). Petroleum ether was used for fat extraction. The crude fiber content was determined by boiling the sample in a mixture of acetic, nitric (V), and trichloroacetic acids. The resulting precipitate was washed on the filter and dried at 105 °C to constant weight. The filter was ashed in a furnace at a temperature of 550 °C and weighed. The crude fiber content was calculated from the mass difference. 

#### 2.2.2. Measurement of Thermodynamic Characteristics of Gelatinization

The thermal properties of wheat flour (control) and wheat flour in the presence of nuts oil cakes were measured using a DSC 4000 differential scanning calorimeter (PerkinElmer, Waltham, MA, USA) [13,18,19]. For this purpose, 3.5 mg (dry weight basis, dwb) of the sample was weighed into an aluminum pan, and then 10.5 mg of distilled water was added to it. The pan was hermetically sealed and left at room temperature for 24 h for moistening. Subsequently, the samples were heated in the temperature range of 30–100 °C with a heating rate of 10 °C/min. A reference sample was an empty aluminum pan. On the basis of the obtained thermograms, the value of the onset temperature T_O_, the peak T_P_, the end temperature T_E_ (°C), and the gelatinization enthalpy ΔH_G_ (J/g) were determined. The difference between the T_P_ and T_E_ (ΔT; °C) was also calculated. The cooling samples were stored at 5 ± 1 °C for 7 days and scanned in the same way by the calorimeter. The onset of (T_O_), the peak (T_P_), the end (T_E_) transition temperatures, and the value enthalpy of melting of the recrystallized amylopectin ΔH_R_ (J/g) were determined. Additionally, the percentage of retrogradation (R) (ΔH_R_/ΔH_G_) × 100 was computed. 

#### 2.2.3. Measurement of Pasting Characteristics

The pasting properties were performed by an RVA (Rapid Visco Analyzer, Tec Master, Perten Instruments, Hägersten, Sweden). For this purpose, 10% (*w*/*w*) water suspensions of all samples were prepared. The appropriate amount of sample and water were weighed in the aluminum vessel of the RVA analyzer. During the measurement, the sample was stirred (160 rpm) and heated for 1 min at 50 °C. Heating was continued to 95 °C at a rate of 12 °C/min, and the sample was held at this temperature for 5 min. It was then cooled to 50 °C at a rate of 12 °C/min and finally held at 50 °C for 2 min. During the measurement, the viscograms were obtained, from which the following parameters were read: pasting temperature (PT) (°C), maximum viscosity (PV) (mPa·s), viscosity at 95 °C (HPV) (mPa·s), final viscosity at 50 °C (FV) (mPa·s), viscosity decrease on heating (BD) (mPa·s), and viscosity increase on cooling (SB) (mPa·s) [13,19,20].

#### 2.2.4. Determination of Viscosity Curves

A rotary viscometer Rheolab QC (Anton-Paar, Graz, Austria) equipped with a system of coaxial cylinders was used to determine the viscosity curves [19]. The paste resulting from the RVA measurement (as described in Section 2.2.3) was placed into the measuring element of the rheometer and subjected to shearing at subsequently increasing and decreasing shear rate from the range of 1–300 s^−1^ at 50 °C. The resulting viscosity curve determined in the range of increasing shear rate was described by the power law model:(1)$$\eta_{ap}=K\cdot {\dot{\gamma }}^{n-1}$$
where *η_ap_*—apparent viscosity [Pa·s], *K*—consistency coefficient [Pa·s^n^], $\dot{\gamma }$—shear rate [s^−^^1^], and *n*—flow behavior index [19].

The hysteresis loop area (Pa s^−1^) between the flow curves determined in the range of increasing and decreasing shear rate was calculated in order to assess the thixotropic properties of the examined paste.

#### 2.2.5. Determination of Mechanical Spectra

The rheometer (MARS II, Thermo Fisher Scientific, Waltham, MA, USA) with a system of parallel plates (diameter 35 mm, gap size 1 mm) was used to determine the viscoelastic properties of examined pastes (from an RVA analyzer). The properties of the analyzed systems were characterized at a temperature of 25 °C. The paste samples were placed in the rheometer system and left for 2 min. Then, the mechanical spectra were determined in the range of linear viscoelasticity at a constant strain amplitude of 0.1% in the angular frequency range of 1–100 rad/s. The resulting mechanical spectra were described by the power law models [13,18,19]:(2)$$G^{\prime }\left(\omega \right)=K\prime \cdot \omega^{n\prime }$$
(3)$$G^{\prime \prime }\left(\omega \right)=K^{\prime \prime }\cdot \omega^{n^{\prime \prime }}$$
where *G′*—storage modulus (Pa), *G″*—loss modulus (Pa), *ω*—angular frequency (rad/s), and *K′*,* K″*, *n′*, and *n″*—experimental constants.

Based on the values of both moduli, the tangent of the phase shift angle (tan *δ* = G″/G′) was calculated, and its dependence on the angular frequency (*ω*) was plotted for all of the systems.

#### 2.2.6. Statistical Analysis

The statistical analyses were performed using the Statistica 13.3 program (StatSoft, Cracow, Poland). All of the experiments were performed in triplicate, and the results are presented as the mean ± standard deviation (SD). In order to evaluate the significance of the difference between the mean values of the parameters determined, two-way analysis of variance (ANOVA) and Duncan’s test were used at a significance level of 0.05. A two-way ANOVA test was performed to evaluate whether the type of oil cakes (factor 1), the level of substitution (factor 2) and if there are interactions between these two factors (factor 1 × factor 2) that affect the obtained results. Additionally, the values of Pearson′s linear correlation coefficients were calculated between the analyzed parameters and their significance was tested at the significance level of 0.05.

## 3. Results and Discussion

### 3.1. Chemical Composition

The content of selected chemical components in the tested systems containing nut oil cakes is listed in Table 1. The tested systems did not differ significantly in terms of moisture, and the value of this parameter was, on average, 13.8%. Samples containing nut oil cakes had a higher fat content compared with WF (control). The fat content in these systems was low but increased significantly as the share of oil cakes increased. It was shown that the average content of this parameter in the case of a system with 15% cake content was 4.4%. After cold pressing, a small amount of this ingredient remains in the oil cake. Hence its presence in the systems. According to Bochkarev et al. [21] and Barta et al. [11], the tolerated level of fat content in oilseed cakes is approximately 25%. Above this value, there is a risk of rapid fat oxidation and, at the same time, difficulties in mixing such raw material with other ingredients. The content of minerals and crude fiber in systems containing peanut cakes increased with their increasing share. Oilseed cakes are a very good source of minerals and fiber. Barta et al. [11] reported that the content of minerals in oil cake flour based on sunflower, pumpkin, or poppy seeds was 5.99%, 9.14%, and 10.99%, respectively. This indicates the beneficial health profile of cakes and the need to use them in the composition of food products. The use of hazelnut or walnut cake as a functional ingredient for baking bread will improve the nutritional value and health benefits of this category of food. Moreover, according to Sivaramakrishnan and Gangadharan [22], cakes of various botanical origins are used in biotechnological processes to obtain enzymes, secondary metabolites, or biomass.

### 3.2. Gelatinization and Retrogradation Properties of Tested Samples

Table 2 contains the values of parameters illustrating the gelatinization process of wheat flour and its systems with nut cakes, determined using differential scanning calorimetry (DSC). DSC is a technique that shows the transition of starch from its crystalline state when heated in an aqueous environment. As part of the analysis, parameters such as T_O_, T_P_, T_E_, and ΔH_G_, ΔH_R_ were determined for the gelatinization and retrogradation processes. The gelatinization process is characterized by the melting of pseudocrystalline areas of amylopectin, which is accompanied by limited elution of amylose and loss of microscopic birefringence. This phenomenon can be visualized under a microscope in polarized light [23]. T_O_ is the temperature at which the gelatinization process begins. This parameter also determines the melting temperature of crystals in starch granules. In turn, the T_P_ parameter or peak temperature determines the endothermic peak in the DSC thermogram and is a measure of the quality of the crystallite, i.e., the length of the double helix [24], whereas T_E_, or final temperature, is the final temperature at which the sample is completely gelled. The ΔH_G_ parameter or enthalpy of transformation or gelatinization enthalpy (J/g) is calculated on the basis of the DSC endotherm, expressing the energy needed to destroy the double helix structure during starch gelatinization [25].

The values of the tested T_O_, T_P_, and T_E_ parameters ranged from 58.8–60.2 °C, 64.1–65.5 °C, and 69.8–71.1 °C, respectively. The value of T_P_, i.e., the temperature at the beginning of the phase transformation, is a measure of the quality of the crystal structure and reflects the length of the double helix [24,26]. In the case of the pasting process, it was found that wheat flour and flour with 5% HOC content did not differ statistically significantly in these parameters. The conducted two-factor analysis of variance showed that the value of the T_O_ parameter was significantly influenced only by the level of oil cake substitution, and the value of the T_P_ parameter was also influenced by the interaction between the type of oil cake and the level of its substitution. It was shown that none of the tested factors influenced the value of the T_E_ parameter, and the tested samples did not differ significantly in this parameter. The gelatinization enthalpy ΔH_G_ ranged from 4.3 J/g (WFHOC5%) to 5.9 J/g (WFWOC5%). This parameter determines the crystallinity of starch granules in terms of quantity and quality and is a measure of the loss of molecular order in the molecule [24,26]. According to the cited authors, the size distribution of starch grains, their shape and the presence of phosphate esters have a significant impact on the value of the gelatinization enthalpy. In the case of flour and its systems with non-starch ingredients, the ingredient undergoing the gelatinization process is starch. According to Singh et al. [24] and Gałkowska and Juszczak [18], the transition temperatures and enthalpy of starch gelatinization characteristic of this process are a function of several factors. The most important are the degree and molecular architecture of crystallinity, the molecular structure of amylopectin, the composition of starch, the shape of the grains, and their size distribution. According to Noda et al. [27], the value of DSC parameters depends on the molecular structure of amorphous regions, which is related to the distribution of shorter chains in amylopectin, which plays a key role in the crystallinity of starch granules [24]. According to Singh et al. [24], the characteristic phase transition temperatures measured using DSC reflect the properties of starch granules, such as the degree of crystallinity. This depends on the chemical composition of the starch and is helpful in determining the thermal and rheological properties of the starch. This is because starches of different botanical origins, included in different flours, have different phase transition temperature ranges and differ in the values of gelatinization or retrogradation enthalpies [24]. Increased phase transition temperatures result from a high degree of crystallinity, which provides greater stability and complicates the gelatinization process [23]. Tester et al. [28] claim that the degree of perfect crystallinity is reflected in the gelatinization temperature of starch or systems in which starch occurs. According to Tester [28] and Singh et al. [24], the process of swelling of starch grains and the process of their subsequent gelatinization is a function of many factors, such as the structure of amylopectin, i.e., the molecular weight of this fraction, the degree of branching of the structure and its polydispersity, the chemical composition of starch (the ratio of amylose to amylopectin, the content of non-starch components, mainly phosphorus) and the grain structure, i.e., the ratio of amorphous to crystalline areas. Characteristic parameters describing the retrogradation process of starch contained in the tested systems were also determined using the DSC method. During cooling, it is observed that the starch chains gradually retrograde, forming partially ordered structures. The retrogradation process initially involves a rapid recrystallization of the amylose fraction, followed by a slow recrystallization of the amylopectin molecules. The determined DSC endotherms of retrograded starch enable quantitative measurement of enthalpy changes and transition temperatures to the melting of crystallites [18]. The tested samples differed slightly in terms of the described parameters. The conducted two-factor analysis of variance showed that the value of the T_O_ parameter was significantly influenced only by the level of oil cake addition and the interactions of both factors, while the value of the T_P_ parameter was influenced only by factor 2 (Table 2). However, the value of ΔH_R_ depended significantly on the type of oil cake and the level of its addition to WF. It was shown that the values of parameters illustrating the retrogradation process were, in all cases, lower compared with analogous parameters illustrating the gelatinization process. This is consistent with the observations of Singh and colleagues [24]. Cited authors claim that the enthalpy of starch retrogradation is usually lower by an average of 60–80% compared with the enthalpy of gelatinization. However, the characteristic transition temperatures are, on average, 10–26 °C lower than the temperatures illustrating the gelatinization process. According to Gałkowska and Juszczak [18], this is due to the fact that the reconnection of amylopectin chains is random, and the degree of crystallinity will not be the same as before gelatinization. The value of the ΔH_R_ parameter for both types of cakes decreased significantly as their share increased. The opposite trend was found by Pycia and Juszczak [13], who observed an increase in the value of this parameter in samples in which WF was substituted with ground hazelnuts (H) and walnuts (W) in the same proportions as in the presented work. This is probably related to the higher fat content in these systems compared with those examined in this work. The calculated degree of retrogradation (R) for the control sample was 24%, and the addition of peanut cakes resulted in a significant reduction in the value of this parameter and a higher share of peanut cakes in the system. The opposite tendency was observed by Pycia and Juszczak [13]. According to the cited authors, the thermal properties of starch in systems containing a non-starch component depend on its type and content but also on the size and shape of the starch grain or the length and degree of branching of amylopectin.

The research found, among others, a significant positive linear correlation between the values of parameters illustrating the gelatinization T_O_ process and the values of T_P_, ΔH_R_, and R parameters and the content of ash, fat, and protein (r = 0.85, r = −0.91, r = −0.94, r = 0.76, r = 0.84, and r = 0.83, respectively; *p* < 0.05).

### 3.3. Pasting Properties

In the presence of water and at the right temperature, starch grains absorb water and swell and thus increase their volume several times compared with their original size. Next, the starch grain breaks and amylose flows out, forming a three-dimensional network. Starch suspensions have different viscosities depending on factors such as concentration, temperature and mechanical forces. Changes in starch viscosity during the gelatinization process can be monitored on the basis of the gelatinization curves plotted during the RVA analysis [23]. Among the experimental parameters, the temperature value and heating time of the tested suspensions are important in shaping the rheological characteristics, including gelatinization characteristics. Pasting characteristic curves are shown in Figure 1 and Figure 2. In turn, the values of pasting characteristic parameters determined on the basis of the curves are listed in Table 3. Based on the determined pasting curves, it was found that the presence of hazelnut or walnut cake in the tested systems had a statistically significant influence on their course. It was shown that in the case of both types of cakes, with their increased share, the maximum and final viscosity of these systems decreased significantly. This is consistent with the gelatinization characteristics of systems determined by Pycia and Juszczak [13], in which WF was substituted with H or W in the amounts of 5%, 10%, and 15%. According to the authors [18,24], the maximum viscosity of systems containing starch at a given concentration reflects the swelling potential of the starch granules freely before disintegrating under elevated temperature conditions; therefore, it should be assumed that in the tested systems, the presence of nut cakes not containing starch hindered the swelling of starch grains. Moreover, the increasing share of the non-starch component in the system and the decreasing share of starch responsible for the viscosity of the system resulted in a visible decrease in the viscosity of such a system. This is consistent with previous observations, where the influence of different additions of walnuts and hazelnuts on the rheological properties of wheat flour was investigated [13]. Moreover, the analysis of the pasting curves shows that keeping the tested pastes at a temperature of 95 °C for several minutes resulted in a significant reduction in the paste viscosity, which clearly indicates low thermal stability and low resistance of these pastes to increased temperature and shear force. Therefore, it should be assumed that dough based on such systems will also have modified rheological properties. The rheological system containing starch exhibits different viscosities depending on concentration, temperature, and shear rate [24].

Based on the results of the two-way analysis of variance, it was found that, in general, the type of nut oil cakes, the level of substitutions and the interaction between these two factors had a significant impact on the gelatinization parameters of the systems tested in this study. It was shown that the control sample, i.e., wheat flour, gelatinized at the lowest temperature (Table 3). However, samples containing nut cake gelatinized at a significantly higher temperature compared with WF (by 1.2 °C on average), but the differences between the average values were not significant. The slight increase in PT probably indicates that the presence of nut cakes in the mixture with flour could have hindered the gelatinization of the starch contained therein. Pycia and Juszczak [13] also observed an increase in the PT value in systems where the share of H or W increased. Pasting temperature is a measure of the ability of starch granules to absorb water and swell. PT is the lower, the easier and faster the grains absorb water and increase their volume [29]. The key parameter demonstrating the rheological properties of the preparation is maximum viscosity (PV). It was found that this parameter depended significantly on all tested factors. Replacing part of the flour with walnut oil cake significantly reduced the maximum viscosity of the gruel, which is consistent with previous observations related to the replacement of wheat flour with H or W [13]. The mean PV of WFHOC samples was 7% higher than WFWOC but 16% lower than WF. In the case of both systems, PV decreased as the OC share increased. The lower viscosity of WF and OC systems can probably be explained by the difficult process of starch gelatinization and the reduced share of starch in the system due to the partial replacement of flour with nut cakes. The HPV parameter shows the viscosity of the pastes at a temperature of 95 °C. It was found that the control sample had the highest value of this parameter, and the WFWOC15% sample had the lowest. WFWOC systems had lower HPV values compared with WFHOC. In both types of systems, the value of this parameter decreased significantly as the number of cakes in the system increased. The decrease in viscosity in relation to PV, noticeable in the course of the gelatinization curves, during holding at a temperature of 95 °C is illustrated by the BD parameter. The value of this parameter ranged from 610 mPa·s (WFWOC) to 875 mPa·s (control) (Table 3). WFHOC samples had a 10% greater decrease in this parameter compared with WFWOC. The BD value of systems with cakes decreased as their share increased. This proves the increasing stability of the pastes under the influence of the increasing share of HOC and WOC compared with the control sample. The stability of the paste at elevated temperatures and under the influence of mechanical forces is an important technological parameter. Therefore, it can be concluded that the presence of nut cakes in these systems improves their stability. Therefore, the structure of the tested WF/OC-based pastes became more resistant to shear forces and high temperatures compared with WF. It was also shown that the value of the SB parameter was lower for samples with cakes compared with the control sample and decreased significantly as their share in the system increased. This proves that the rheological stability of WFHOC and WFWOC pastes increases during cooling and storage at 50 °C. This is consistent with observations regarding the influence of ground hazelnuts and walnuts on the properties of wheat flour [13]. Therefore, the rheological stability of the pastes of systems containing peanut cakes increased (lower SB values) in the cooling phase and kept at 50 °C. The FV parameter shows the final viscosity of the paste. Both the course of the pasting curves (Figure 1 and Figure 2) and the value of the FV parameter confirm that cooling the paste to a temperature of 50 °C resulted in an increase in viscosity in relation to the values determined during holding at a temperature of 95 °C. The control sample had the greatest increase in viscosity after cooling the paste, even exceeding PV (Table 3). Higher FV values compared with PV were also recorded in the case of WFHOC5% and 10% samples. In the remaining cases, FV was lower than PV. Schmiele et al. [30] analyzed the gelatinization characteristics of mixtures of wheat flour with wheat bran and whole grain wheat flour and also found a decrease in the values of the PV, BD, and SB parameters as the share of these ingredients increased. The cited authors explained this tendency by the dilution of starch contained in flour (its decreasing share) and the increasing share of fiber, fat, and minerals.

The research found, among others, a significant positive linear correlation between the values of the PT parameter and the values of parameters showing the retrogradation process T_O_, T_P_, ΔH_R_, PV, and BD and the fat and protein content (r = 0.85, r = 0.92, r = −0.78, r = −0.76, r = −0.76, r = 0.75, and r = 0.82, respectively; *p* < 0.05). Moreover, a linear correlation was demonstrated between the PV parameter and parameters illustrating the T_O_ gelatinization process, T_O_ retrogradation, T_P_, ΔH_R_, R, ash, fat, and protein content (r = −0.84, r = −0.86, r = −0.78, r = 0.94, r = 0.85, r = −0.75, r = −0.94, and r = −0.87, respectively; *p* < 0.05).

### 3.4. Flow Behavior and Thixotropy

Examples of viscosity curves and flow curves with thixotropy hysteresis loops of the tested samples are shown in Figure 3 and Figure 4, respectively. The parameters of the power equation used to describe the viscosity curves are summarized in Table 4. The course of the viscosity curves indicates the influence of the addition of nut oil cakes and the amount of their share in the systems on the viscosity of pastes and other rheological features related to flow.

It was shown that the pastes of all tested samples showed the characteristics of a non-Newtonian, shear-thinning fluid and the phenomenon of thixotropy (Figure 4). Therefore, in all cases, the apparent viscosity decreased with increasing shear rate. It was also shown that the presence of nut oil cakes in WF systems resulted in a significant reduction in their apparent viscosity. The greater, the higher the OC content (Figure 3). This is confirmed by the parameter values of the power model used to describe the experimental curves. It was shown that the WF had the highest value of the consistency coefficient, and the value of this parameter statistically decreased significantly as the share of OC in the systems increased. The average value of the K parameter of WFWOC systems was 29% lower compared with the control sample and 16% lower than WFHOC. A two-way analysis of variance showed that the value of this parameter depended on all tested factors (*p* ≤ 0.001). This is consistent with the determined gelatinization characteristics, in which a decrease in PV was observed as the OC content increased. The tests revealed slight statistical differences in the samples in terms of the flow index (n). The WFWOC15% sample had the lowest value of this parameter, which indicates the greatest decrease in viscosity under the influence of shear forces. Both the consistency index and the flow index are related to the amylose content of the starch because Pycia et al. [20] showed a positive correlation between the amylose content in potato starch and the consistency coefficient values, as well as a negative correlation between the amylose content and the melt flow index values. Therefore, in systems in which part of the flour was replaced with a non-starch factor, the total amylose content also decreased, which may be consistent with the determined correlation.

Figure 4 shows exemplary flow curves based on which the area of the hysteresis loop (HA) was determined. The area between the upper and lower flow curves is used to quantify the thixotropy phenomenon. All tested samples showed the phenomenon of thixotropy, which reflects the time necessary to transition from a given microstructural state to another [31]. Pastes containing WFHOC were characterized by higher values of the hysteresis loop areas compared with WF and WFWOC (Figure 5). This indicates that these systems needed more time to rebuild their structure after the applied stress was removed. A linear correlation was demonstrated between the K parameter and parameters illustrating the T_O_ gelatinization process, T_O_ retrogradation, T_P_, ΔH_R_, R, ash, fat, and protein content (r = −0.84, r = −0.86, r = −0.78, r = 0.94, r = 0.85, r = −0.75, r = −0.94, and r = −0.87, respectively; *p* < 0.05)

### 3.5. Viscoelastic Properties

The mechanical properties of the tested samples were assessed using the sweep frequency test, as the dependence of the storage moduli G′ and loss modulus G″ on the angular velocity. The first one, G′, is a measure of the energy stored in the material during the cycle, and the loss modulus G″ reflects the energy lost and dissipated during the sinusoidal deformation cycle. In turn, the ratio of energy lost to stored in the cycle is determined by the parameter tangent of the phase shift angle (tan δ = G″/G′) [24]. Figure 6 shows the mechanical spectra of the control sample (WF) and the tested WFHOC and WFWOC systems, which are the dependence of the G′ modulus and the loss modulus G″ on the angular velocity. However, Figure 7 shows the values of the phase shift angle tan δ (tan δ = G″/G′). It was found that for all analyzed samples, the values of the storage modulus (G′) were higher than the values of the loss modulus (G″). This clearly proves the dominance of elastic features over viscous features. In all analyzed cases, a constant increase in the values of the storage modulus and loss modulus with angular velocity was also observed. The values of the phase shift angle tan δ calculated in the angular frequency range 1–100 rad·s^−1^ were also in all cases smaller than unity and greater than 0.1, which additionally confirms the dominance of elastic over viscous features and indicates that the tested systems have nature of weak gels. In the case of WFHOC, the values of the G′ and G″ modulus were very close to each other. However, in the case of WFWOC, the values of G′ and G″ differed depending on the level of WOC in the system. It should be stated that the lowest values were recorded in the samples in which WF was replaced by WOC, comparing the values of the storage modulus and the loss modulus in the entire range of angular velocity determined experimentally for all the analyzed systems. This is confirmed by the previously presented results related to gelatinization curves and apparent viscosity. These observations are consistent with the results presented by Pycia and Juszczak [13], where the impact of substituting wheat flour with ground walnuts or hazelnuts was analyzed.

Table 5 shows the parameter values of the power equations used to describe the determined mechanical spectra. The conducted two-factor analysis of variance showed the influence of most of the tested factors on the values of these parameters. It was found that the values of the K′ and K″ parameters indicating the initial values of the storage modulus and loss modulus were significantly higher in the case of the control sample and WFHOC samples compared with the WFWOC sample. However, no significant differences were found between the control sample and WFHOC samples in terms of these parameters. Replacing WF with walnut oil cakes resulted in a significant reduction in the value of the K′ parameter, which indicates a weakening of the elastic properties, and this phenomenon increased as the share of WOC in the system increased. A similar trend was observed for the K″ value. These observations mirror similar changes observed in gelatinization characteristics. Different behaviors of the systems depend on the type of cakes in the system and their level. The tested samples differed slightly in terms of the values of the parameters n′ and n″, indicating the dependence of the modules on the angular velocity. 

Based on the values of these parameters, the nature of the gel can be determined. Because n′ values close to 0 indicate that the gel is flexible in physical or chemical terms (due to covalent bonds). However, n′ > 0 means that the gel is weak [18]. The statistical analysis performed showed a linear correlation between the K′ parameter and the parameters K, HA, HPV, FV, SB, K″, and n” (r = 0.77, r = 0.86, r = 0.94, r = 0.93, r = 0.91, r = 0.98, r = −0.94; *p* < 0.05). In turn, the K″ parameter correlated, among others, with the HA, HPV, FV, and SB parameters, and the K′ and n” parameters (r = 0.89, r = 0.91, r = 0.90, r = 0.88, r = 0.98, r = −0.97, respectively; *p* < 0.05).

## 4. Conclusions

The study examined the possibilities of using by-products from oil pressing, i.e., hazelnut oil cake (HOC) and walnut oil cake (WOC), in terms of their interaction with wheat flour. It has been shown that oil cakes (OC), as a non-starch ingredient, significantly influence the thermal and rheological properties of flour systems in which wheat starch is the dominant ingredient. So, it was found that the type of oil cakes and the level of their addition influenced the thermal and rheological properties of the tested systems. It has been shown that systems with nut oil cakes were characterized by a higher gelatinization temperature but significantly lower maximum viscosity and increased rheological stability. Moreover, the presence of OC in the systems weakened the viscoelastic properties of the pastes. The percentage of nut oil cakes in the tested systems had a significant impact on the tested properties. It was shown that WFHOC/WFWOC15% systems had reduced viscosity and weakened viscoelastic properties compared with systems with a lower OC content, which is not a favorable feature from the technological point of view. However, these systems were the most stable, which is an advantageous feature. On the other hand, for baking purposes, it is important to know the properties of dough made from such mixtures. It is expected that the dough produced based on these mixtures will probably have changed rheological properties, and the direction of these changes is currently unknown. Therefore, this requires continued research into the rheological properties of dough based on such mixtures. It is also necessary to examine the role that nut oil cakes will play as a functional additive in the process of bread enrichment.

## Figures and Tables

**Figure 1 foods-12-04060-f001:**
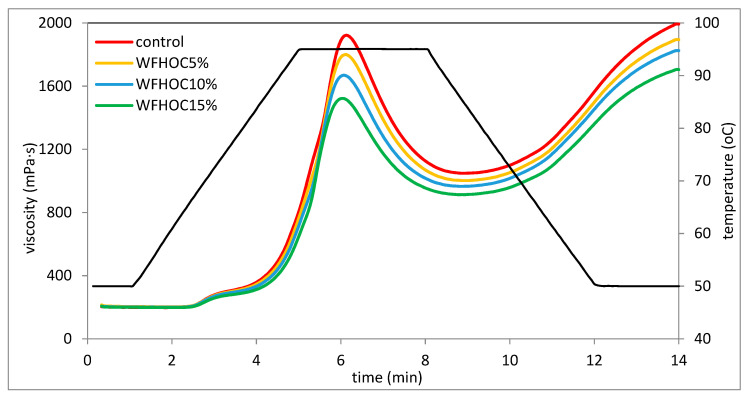
Pasting curves of WF sample and samples with HOC.

**Figure 2 foods-12-04060-f002:**
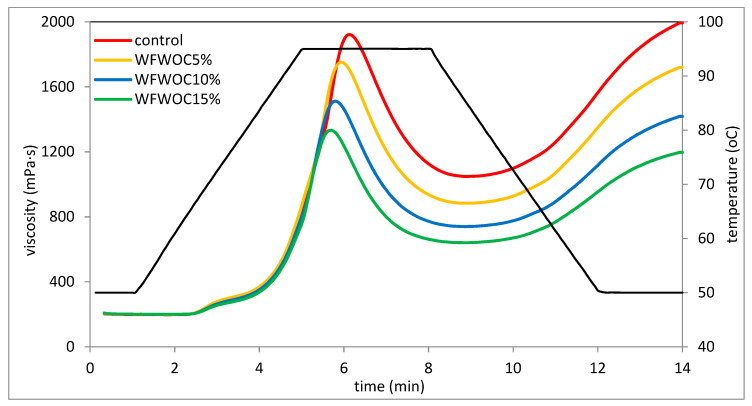
Pasting curves of WF sample and samples with WOC.

**Figure 3 foods-12-04060-f003:**
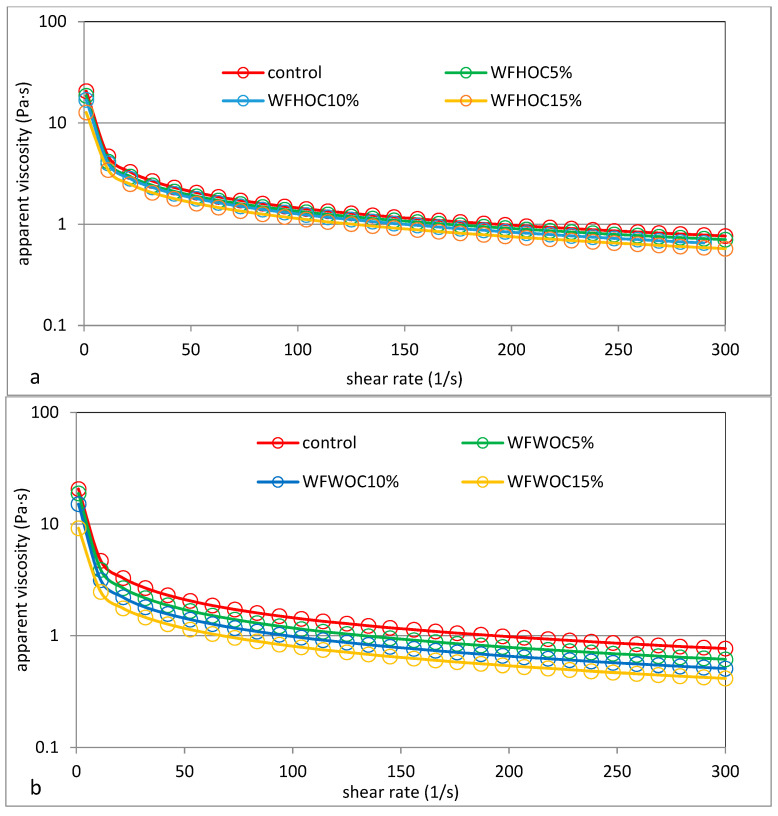
Viscosity curves of wheat flour (control) and system of wheat flour with hazelnut oil cakes (**a**) and walnut oil cakes (**b**).

**Figure 4 foods-12-04060-f004:**
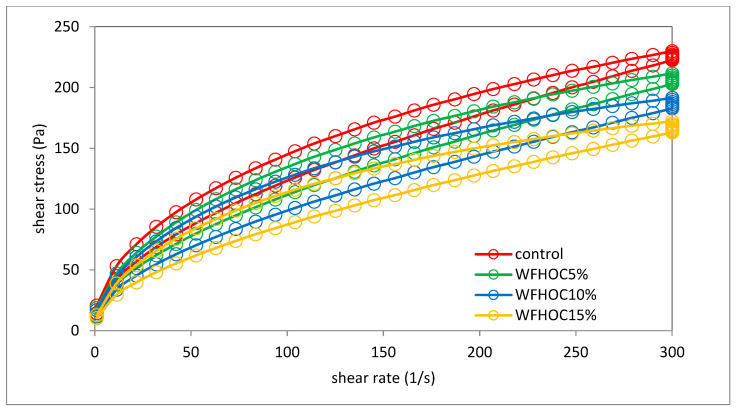
Flow curves of examples samples.

**Figure 5 foods-12-04060-f005:**
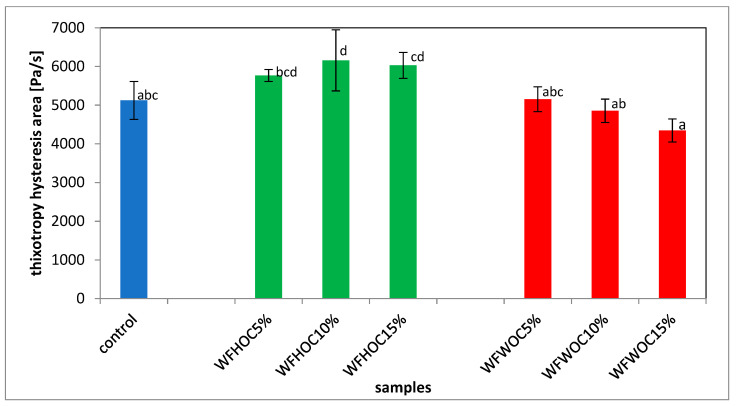
Hysteresis area (HA) of thixotropy of wheat flour (control) and system of wheat flour with hazelnut oil cake and walnut oil cake. Means with different letters in the columns indicate statistically significant differences between samples according to Duncan’s test at significance level of α = 0.05.

**Figure 6 foods-12-04060-f006:**
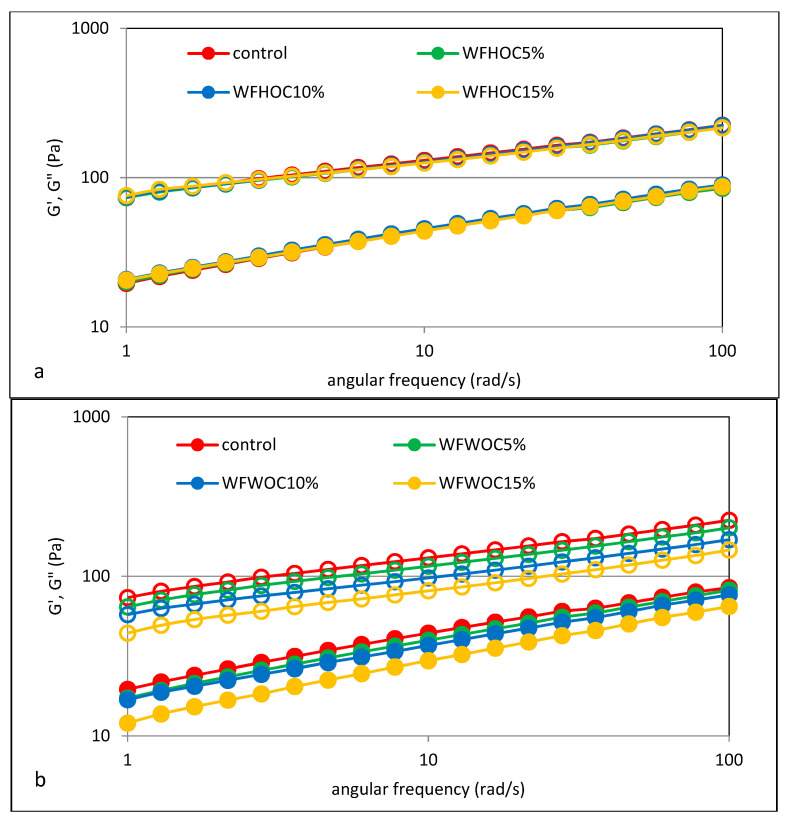
Mechanical spectra of wheat flour and system of wheat flour with hazelnut (**a**) and walnut oil cake (**b**). G′—empty markers; G″—filled markers.

**Figure 7 foods-12-04060-f007:**
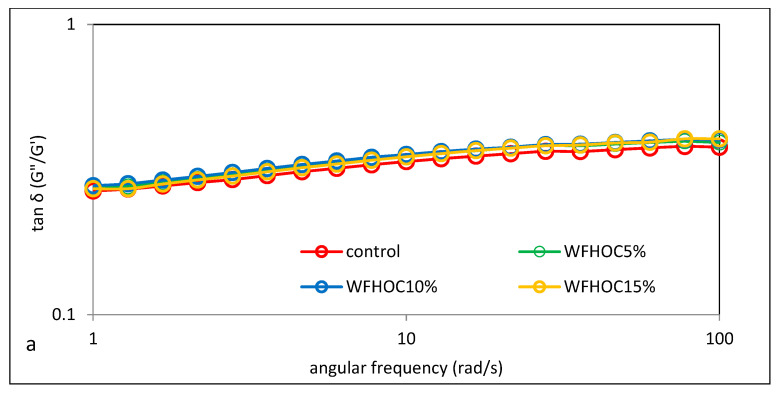
Tangent δ depending on the angular stress of wheat flour (control) and system of wheat flour with hazelnut oil cake (**a**) and walnut oil cake (**b**).

**Table 1 foods-12-04060-t001:** Selected chemical composition of the tested systems with nut oil cakes.

Sample	Moisture Content [%]	Lipid Content [%]	Ash Content [%]	Crude Fiber Content [%]
Control	14.0 ^a^ ± 0.1	0.9 ^a^ ± 0.1	0.7 ^a^ ± 0,1	2.2 ^a^ ± 0.0
WFHOC5%	14.1 ^a^ ± 0.0	1.7 ^b^ ± 0.0	0.7 ^a^ ± 0.1	7.4 ^b^ ± 0.2
WFHOC10%	14.0 ^a^ ± 0.1	2.6 ^c^ ± 0.1	1.1 ^b^ ± 0.2	9.3 ^d^ ± 0.6
WFHOC15%	13.5 ^a^ ± 0.1	4.3 ^d^ ± 0.7	1.3 ^b^ ± 0.1	12.0 ^e^ ± 0.2
WFWOC5%	13.8 ^a^ ± 0.0	1.6 ^b^ ± 0.1	0.7 ^a^ ± 0.0	8.3 ^c^ ± 0.1
WFWOC10%	13.6 ^a^ ± 0.1	3.1 ^c^ ± 0.6	0.8 ^a^ ± 0.1	8.2 ^c^ ± 0.2
WFWOC15%	13.5 ^a^ ± 0.0	4.5 ^d^ ± 0.3	1.2 ^b^ ± 0.1	12.9 ^e^ ± 0.3

Results are expressed as mean values ± standard deviations (SD). Means with different letters in the columns indicate statistically significant differences between samples according to Duncan’s test at significance level of α = 0.05.

**Table 2 foods-12-04060-t002:** Thermodynamic characteristics of gelatinization and retrogradation of systems wheat flour with hazelnut and walnut oil cakes.

Sample	Gelatinization					Retrogradation					
	T_O_ (°C)	T_P_ (°C)	T_E_ (°C)	ΔT (°C)	ΔH_G_ (J/g)	T_O_ (°C)	T_P_ (°C)	T_E_ (°C)	ΔT (°C)	ΔH_R_ (J/g)	R (%)
Control	58.8 ^a^ ± 0.9	64.2 ^a^ ± 0.8	71.1 ^a^ ± 0.4	12.3 ^a^ ± 1.2	5.3 ^d^ ± 0.5	45.8 ^a^ ± 0.9	52.3 ^a^ ± 0.2	62.4 ^ab^ ± 1.3	16.6 ^b^ ± 1.4	1.3 ^d^ ± 0.1	24.0 ^c^ ± 3.8
WFHOC5%	58.9 ^a^ ± 0.4	64.1 ^a^ ± 0.0	70.3 ^a^ ± 0.9	11.4 ^a^ ± 1.3	4.3 ^b^ ± 0.4	48.3 ^bc^ ± 1.8	54.5 ^ab^ ± 2.6	63.3 ^b^ ± 2.2	15.0 ^a^ ± 3.5	0.9 ^c^ ± 0.1	22.3 ^c^ ± 4.1
WFHOC10%	60.1 ^b^ ± 0.3	65.2 ^ab^ ± 0.3	71.3 ^a^ ± 0.6	11.2 ^a^ ± 0.4	4.8 ^bc^ ± 0.1	47.5 ^bc^ ± 0.4	55.7 ^b^ ± 0.9	61.8 ^ab^ ± 0.1	14.3 ^a^ ± 0.4	0.7 ^b^ ± 0.1	14.4 ^b^ ± 1.4
WFHOC15%	60.2 ^b^ ± 0.1	65.5 ^b^ ± 0.2	70.6 ^a^ ± 1.4	10.4 ^a^ ± 1.5	4.7 ^bc^ ± 0.3	49.3 ^cd^ ± 0.8	55.9 ^b^ ± 0.3	62.1 ^ab^ ± 0.8	12.8 ^a^ ± 1.4	0.4 ^a^ ± 0.1	8.9 ^a^ ± 2.9
WFWOC5%	59.6 ^ab^ ± 0.3	65.0 ^ab^ ± 0.4	71.1 ^a^ ± 0.6	11.5 ^a^ ± 0.4	5.9 ^e^ ± 0.4	46.8 ^b^ ± 0.7	52.8 ^a^ ± 0.5	61.7 ^ab^ ± 0.3	14.9 ^a^ ± 0.8	0.7 ^b^ ± 0.0	12.6 ^ab^ ± 1.4
WFWOC10%	60.0 ^b^ ± 0.6	65.5 ^b^ ± 0.4	70.8 ^a^ ± 1.6	10.8 ^a^ ± 1.0	5.1 ^c^ ± 0.4	48.1 ^bc^ ± 0.3	54.5 ^ab^ ± 0.6	61.0 ^a^ ± 1.1	12.9 ^a^ ± 0.9	0.6 ^b^ ± 0.1	11.6 ^ab^ ± 1.7
WFWOC15%	60.1 ^b^ ± 0.7	64.7 ^b^ ± 0.4	69.8 ^a^ ± 0.6	9.7 ^a^ ± 0.2	3.6 ^a^ ± 0.5	50.2 ^d^ ± 1.1	56.1 ^b^ ± 0.5	61.8 ^ab^ ± 0.9	11.6 ^a^ ± 1.5	0.3 ^a^ ± 0.1	9.7 ^ab^ ± 2.8
	Two-way ANOVA *p*-values										
Factor 1	*p* = 0.381	*p* = 0.044	*p* = 0.781	*p* = 0.468	*p* = 0.128	*p* = 0.945	*p* = 0.131	*p* = 0.109	*p* = 0.299	*p* ≤ 0.001	*p* ≤ 0.001
Factor 2	*p* ≤ 0.001	*p* ≤ 0.001	*p* = 0.394	*p* = 0.063	*p* ≤ 0.001	*p* ≤ 0.001	*p* ≤ 0.001	*p* = 0.280	*p* = 0.058	*p* ≤ 0.001	*p* ≤ 0.001
Factor 1 × factor 2	*p* = 0.259	*p* ≤ 0.001	*p* = 0.429	*p* = 0.780	*p* ≤ 0.001	*p* = 0.101	*p* = 0.373	*p* = 0.607	*p* = 0.786	*p* = 0.449	*p* ≤ 0.001

Results are expressed as mean values ± standard deviations (SD). Means with different letters in the columns indicate statistically significant differences between samples according to Duncan’s test at significance level of α = 0.05.

**Table 3 foods-12-04060-t003:** Pasting characteristics of systems wheat flour with hazelnut and walnut oil cake.

Sample	PT (°C)	PV (mPa·s)	HPV (mPa·s)	BD (mPa·s)	FV (mPa·s)	SB mPa·s)
Control	64.8 ^a^ ± 0.1	1923 ^f^ ± 4	1048 ^g^ ± 10	875 ^f^ ± 8	1993 ^f^ ± 6	945 ^g^ ± 6
WFHOC5%	66.1 ^b^ ± 0.8	1804 ^e^ ± 28	1002 ^f^ ± 24	802 ^e^ ± 9	1894 ^e^ ± 28	893 ^f^ ± 8
WFHOC10%	66.2 ^b^ ± 0.9	1670 ^c^ ± 16	965 ^e^ ± 13	705 ^c^ ± 7	1825 ^d^ ± 14	860 ^e^ ± 2
WFHOC15%	66.1 ^b^ ± 0.5	1522 ^b^ ± 6	912 ^d^ ± 12	610 ^a^ ± 10	1705 ^c^ ± 13	792 ^c^ ± 3
WFWOC5%	65.4 ^b^ ± 0.5	1752 ^d^ ± 15	884 ^c^ ± 10	868 ^f^ ± 6	1720 ^c^ ± 13	836 ^d^ ± 3
WFWOC10%	66.1 ^b^ ± 0.4	1512 ^b^ ± 7	740 ^b^ ± 5	772 ^d^ ± 5	1418 ^b^ ± 6	678 ^b^ ± 2
WFWOC15%	66.4 ^b^ ± 0.0	1333 ^a^ ± 11	640 ^a^ ± 9	692 ^b^ ± 3	1197 ^a^ ± 22	557 ^a^ ± 13
Two-way ANOVA *p*-values						
Factor 1	*p* = 0.524	*p* ≤ 0.001	*p* ≤ 0.001	*p* ≤ 0.001	*p* ≤ 0.001	*p* ≤ 0.001
Factor 2	*p* = 0.323	*p* ≤ 0.001	*p* ≤ 0.001	*p* ≤ 0.001	*p* ≤ 0.001	*p* ≤ 0.001
Factor 1 × factor 2	*p* = 0.364	*p* ≤ 0.001	*p * ≤ 0.001	*p* = 0.110	*p * ≤ 0.001	*p* ≤ 0.001

Results are expressed as mean values ± standard deviations (SD). Means with different letters in the columns indicate statistically significant differences between samples according to Duncan’s test at significance level of α = 0.05.

**Table 4 foods-12-04060-t004:** Parameters of power law model determined for viscosity curves of pastes of the wheat flour and wheat flour with hazelnut and walnut oil cakes.

Sample	K (Pa∙s^n^)	n	R^2^
control	19.5 ^f^ ± 0.6	0.43 ^b^ ± 0.01	0.9983
WFHOC5%	18.8 ^e^ ± 0.3	0.43 ^b^ ± 0.01	0.9941
WFHOC10%	16.6 ^c^ ± 0.8	0.44 ^bc^ ± 0.01	0.9955
WFHOC15%	13.7 ^b^ ± 0.8	0.45 ^bc^ ± 0.01	0.9912
WFWOC5%	17.8 ^cd^ ± 1.0	0.41 ^a^ ± 0.01	0.9982
WFWOC10%	14.0 ^b^ ± 0.1	0.42 ^ab^ ± 0.00	0.9980
WFWOC15%	9.4 ^a^ ± 0.6	0.46 ^c^ ± 0.03	0.9947
Two-way ANOVA *p*-values			
Factor 1	*p* ≤ 0.001	*p* = 0.266	
Factor 2	*p* ≤ 0.001	*p* ≤ 0.001	
Factor 1 × factor 2	*p* ≤ 0.001	*p* = 0.182	

Results are expressed as mean values ± standard deviations (SD). Means with different letters in the columns indicate statistically significant differences between samples according to Duncan’s test at significance level of α = 0.05.

**Table 5 foods-12-04060-t005:** Parameters of power law equations describing viscoelastic properties (25 °C) of systems wheat flour with hazelnut and walnut oil cakes.

Sample	K′	n′	R^2^	K″	n″	R^2^
control	76.3 ^d^ ± 3.5	0.23 ^ab^ ± 0.01	0.9983	20.7 ^c^ ± 0.1	0.32 ^b^ ± 0.00	0.9966
WFHOC5%	75.3 ^d^ ± 3.7	0.22 ^a^ ± 0.00	0.9983	21.0 ^d^ ± 2.0	0.31 ^a^ ± 0.01	0.9973
WFHOC10%	75.7 ^d^ ± 1.4	0.23 ^ab^ ± 0.00	0.9989	21.6 ^d^ ± 0.8	0.32 ^b^ ± 0.00	0.9982
WFHOC15%	77.0 ^d^ ± 3.7	0.22 ^a^ ± 0.00	0.9978	21.2 ^d^ ± 0.5	0.31 ^a^ ± 0.00	0.9993
WFWOC5%	67.3 ^c^ ± 0.4	0.24 ^c^ ± 0.00	0.9976	18.1 ^b^ ± 0.3	0.33 ^bc^ ± 0.00	0.9982
WFWOC10%	58.8 ^b^ ± 0.9	0.23 ^ab^ ± 0.01	0.9982	17.3 ^b^ ± 1.4	0.33 ^bc^ ± 0.02	0.9994
WFWOC15%	46.2 ^a^ ± 0.3	0.25 ^d^ ± 0.00	0.9973	12.7 ^a^ ± 0.1	0.36 ^d^ ± 0.00	0.9987
	Two-way ANOVA *p*-values					
Factor 1	*p* ≤ 0.001	*p* ≤ 0.001		*p* ≤ 0.001	*p* ≤ 0.001	
Factor 2	*p* ≤ 0.001	*p* = 0.248		*p* ≤ 0.001	*p* ≤ 0.001	
Factor 1 × factor 2	*p* ≤ 0.001	*p* ≤ 0.001		*p* ≤ 0.001	*p* ≤ 0.001	

Results are expressed as mean values ± standard deviations (SD). Means with different letters in the columns indicate statistically significant differences between samples according to Duncan’s test at significance level of α = 0.05.

## Data Availability

All the data are available within the manuscript.

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
