# Peer review of "Influence of Hazelnut and Walnut Oil Cakes Powder on Thermal and Rheological Properties of Wheat Flour"

_foods, 2023, doi:10.3390/foods12224060_

Round 1

Reviewer 1 Report

Comments and Suggestions for Authors

In abstract provide more specific details instead of using general terms like "tested systems," specify the exact proportions of oil cakes used, such as "systems where 5%, 10%, or 15% of wheat flour was replaced with ground oil cakes based on hazelnuts and walnuts." secondly, Summarize the main findings of the study and their implications. 

Introduction:  Expand on the growing consumer trend towards natural, low-processed foods and their preference for plant-based raw materials. Explain the reasons behind this trend, such as health benefits and the presence of bioactive compounds in oil seeds and nuts.

 Explain the nutritional value and bioactive compounds present in oil cakes, highlighting their potential as functional ingredients in the food industry. Mention their use in improving the nutritional value of baked goods.

Elaborate on the importance of assessing the rheological properties of flour and its mixtures with plant raw materials, such as nut cakes, for the bakery industry. Discuss how this assessment helps predict dough behavior, kneading process, and the quality of the final product.

Add the following one more reference for walnut oil composition and properties 

1870-249X-jmcs-65-04-572.pdf (scielo.org.mx)

Methods and materials:

in viscosity curve determination, what type of parameters was set for determination, and also which type of model was used in software for determination of viscosity.

Briefly write each of composition determination experiments, that will help reader to reproduce their work

rewrite or simplify the pasting properties experiments. the present is hard to understand.

results and discussion:

Chemical composition: Explain the significance of the observed differences in chemical components. How do these findings contribute to the existing knowledge? Are there any practical implications or potential applications? Cite other relevant research papers to support your claims and provide a broader context for the discussion.

Gelatinization and retrogradation properties of tested sample:

Some sentences are too long and complex, making it difficult to follow the information. Break down the information into shorter, more concise sentences to improve readability.

 Explain the meaning of acronyms such as TO, TP, TE, ΔHG, and ΔHR, as well as technical terms like gelatinization, retrogradation, and DSC. This will help readers who are not familiar with the specific terminology.

Pasting properties: fine

Table. 2 data is not in tabulated form, I think authors used JPG file instead of world format, check this and provide clearer table.

Highlight the important changes in figure 2, add arrows to indicates the spot and remove the background lines.

Table 3 must be start from next page collectively.

Figure 5 is not attractive, make it more clear and add significant notation. 

Conclusion: Expand the conclusion section with more of application of the study.

Author Response

Reviewer 1

In abstract provide more specific details instead of using general terms like "tested systems," specify the exact proportions of oil cakes used, such as "systems where 5%, 10%, or 15% of wheat flour was replaced with ground oil cakes based on hazelnuts and walnuts." secondly, Summarize the main findings of the study and their implications. 

Response: The additional information were added to the abstract section.

Introduction:  Expand on the growing consumer trend towards natural, low-processed foods and their preference for plant-based raw materials. Explain the reasons behind this trend, such as health benefits and the presence of bioactive compounds in oil seeds and nuts.

 Explain the nutritional value and bioactive compounds present in oil cakes, highlighting their potential as functional ingredients in the food industry. Mention their use in improving the nutritional value of baked goods.

Elaborate on the importance of assessing the rheological properties of flour and its mixtures with plant raw materials, such as nut cakes, for the bakery industry. Discuss how this assessment helps predict dough behavior, kneading process, and the quality of the final product.

Response: The introduction has been improved

Add the following one more reference for walnut oil composition and properties 

1870-249X-jmcs-65-04-572.pdf (scielo.org.mx)

Response: The additional information and reference were added to the manuscript.

Methods and materials:

in viscosity curve determination, what type of parameters was set for determination, and also which type of model was used in software for determination of viscosity.

Response: the additional information were added to the methods section.

Briefly write each of composition determination experiments, that will help reader to reproduce their work

Response: Additional information on the determination of chemical composition in the tested systems is provided in section 2.2.1.

rewrite or simplify the pasting properties experiments. the present is hard to understand.

Response: The description of the pasting characteristics of the tested samples includes details on sample preparation, methods and speed of its heating, as well as reading and calculating parameters. The description has been corrected and simplified wherever possible.

results and discussion:

Chemical composition: Explain the significance of the observed differences in chemical components. How do these findings contribute to the existing knowledge? Are there any practical implications or potential applications? Cite other relevant research papers to support your claims and provide a broader context for the discussion.

Response: The additional information were added to this section.

Gelatinization and retrogradation properties of tested sample:

Some sentences are too long and complex, making it difficult to follow the information. Break down the information into shorter, more concise sentences to improve readability.

Response: According to the Reviewer's suggestions, some sentences have been shortened.

 Explain the meaning of acronyms such as TO, TP, TE, ΔHG, and ΔHR, as well as technical terms like gelatinization, retrogradation, and DSC. This will help readers who are not familiar with the specific terminology.

Response: The names of these DSC analysis parameter symbols are provided in section 2.2.2. Thermodynamic characteristics of gelatinization

Pasting properties: fine

Response: Thank you for your opinion about this section

Table. 2 data is not in tabulated form, I think authors used JPG file instead of world format, check this and provide clearer table.

Response: The manuscript has been prepared in the format required by the Foods journal. The included tables are editable and have not been assembled into jpg formats.

Highlight the important changes in figure 2, add arrows to indicates the spot and remove the background lines.

Response: Figure 2 shows the pasting curves of wheat flour systems with different contents of walnut cake, determined by the universal RVA method. The course of these curves is characteristic of the pasting process of the starch contained in these systems. Individual samples differ in maximum viscosity, represented by peak size, and other parameters, such as final viscosity. The course of these pasting curves and the differences between them are described in detail in the manuscript. In our opinion, the description of the pasting curves determined by the RVA method is sufficient, and additional symbols on the chart are not necessary. The pasting process is illustrated in this way by other researchers

Table 3 must be start from next page collectively.

Response: Tables should not be shared between pages. The authors will try to arrange the text better.

Figure 5 is not attractive, make it more clear and add significant notation. 

Response: Chart 5 has been corrected. Symbols have been added to indicate significant differences between samples.

Conclusion: Expand the conclusion section with more of application of the study.

Response: The conclusions section was improved

Reviewer 2 Report

Comments and Suggestions for Authors

Lack of novelty.

Author Response

Reviewer 2

Lack of novelty.

Response: The manuscript was carefully analyzed and revised in accordance with the comments of other reviewers. In our opinion, it is difficult to find research results on the analysis of nut oil cakes and their interactions in systems with wheat flour in the literature on the subject. The presence of hazelnut or walnut oil cakes is likely to influence the starch gelatinization process, but also the rheological properties of such systems. This has been demonstrated in work. Nut oil cakes are an example of a by-product of the food industry. Counteracting food waste and looking for ways to manage by-products are problems faced by the food industry. By-products are still a rich source of bioactive substances whose potential should be exploited. Therefore, it is still worth looking for opportunities to use this type of raw materials. However, effectively searching for ways to reuse them requires knowledge of their impact on other food ingredients, e.g. starch. Therefore, the presented research allows for expanding knowledge in this area.

Reviewer 3 Report

Comments and Suggestions for Authors

After reading the manuscript "Influence of hazelnut and walnut oil cakes powder on thermal  and rheological properties of wheat flour", I realized that the manuscript showed in some parts the scientific rigour wanted, but in other parts I have missed it.

The authors have presented critical evaluation only in some paragraphs.

The references are not exactly current, besides some sections can be improved .

Thats why I have written some suggestions below.

L.10-  The objective of the abstract is much clearer than in the Introduction, I ask you to re-evaluate it.

L.18-19 - " it was found that the type of oil cakes and the level of their addition influenced the thermal and rheological properties of the tested systems" -  This important statement is not in the conclusion of the paper.

Please, consider my suggestion and re-evaluate your conclusion as well.

L.28- What was the best oil ? hazelnut or walnut ?

What was the best treatment 5%, 10% 15 % ? This needs to appear in the abstract and conclusion, guys.

I missed paragraphs in the text.

L. 36- "The first ones mentioned" - Be clearer.

L.43- All these statements are from the same author ? Strange.... since they are about different subjects. Please, improve this part, it is really confused.

L.66- Authors, it seems to me that you should first present "hazelnut and walnut", then mention both oils and their health benefits and the approach to sustainability. As it is, it's too much back and forth ... and then close with cake.

Would it really be by-product? What do you think of a co-product? Evaluate the real situation of the product used and the definitions.

L.82- "Pycia and Juszczak" - These authors are 11 e not 12.  

L.95- A lot of important information about the preparation of the cake has not been included.

Please, add the detailed and complete formulation of all the treatments, preferably in a table to make it easier for other researchers/readers to understand. Eg : Milk ?sugar ? Whole milk ? demerara sugar ? Author details, please. The formulation must be replicable.

What type of cake was made?  Angel cake ? Cupcake and so on...

Baking time ? Which oven was used ? How many batches ? How many replications ?

L.138- "Frequency sweep " - You used this terminology in this section and then no more. I ask you to re-evaluate all the nomenclatures used in the Material and methods and those used in the results and discussion.

L.154/ 155- Between or among ?

L.317- Figure 1 and 2 need some thoughts and adjustments. The black line should also be in the caption, in my opinion. In all the results in the previous tables, you compared all the treatments. Why did you separate them in this result? I'm sorry, but I don't understand.

L.325- Between ? Check English

L.399-  Same suggestion. In all the results in the previous tables, you compared all the treatments. Why did you separate them in this result? I'm sorry, but I don't understand.

L.451- Follow the same reasoning, at least control, hazelnuts and walnuts with different colors.

L.424- Please, could you confirm the results 0.41a±0.01 ; 0.42b±0.00 ; 0.43ab±0.01

L.484-  Same suggestion. In all the results in the previous tables, you compared all the treatments. Why did you separate them in this result? 

L.530 - I missed the limitations of the study.

It was not good to end with the table before the conclusion.

L.540 - " It is expected that the dough produced based on these mixtures will also have changed rheological properties" -  But this is the conclusion of your paper. It should be a solid sentence.

L.563 - Check Authors Guide about references. Eg  (:) DOI

Comments on the Quality of English Language

Minor editing of English language required

Author Response

Reviewer 3

After reading the manuscript "Influence of hazelnut and walnut oil cakes powder on thermal  and rheological properties of wheat flour", I realized that the manuscript showed in some parts the scientific rigour wanted, but in other parts I have missed it.

The authors have presented critical evaluation only in some paragraphs.

The references are not exactly current, besides some sections can be improved .

Thats why I have written some suggestions below.

L.10-  The objective of the abstract is much clearer than in the Introduction, I ask you to re-evaluate it.

Response: the aim of work in introduction has been corrected

L.18-19 - " it was found that the type of oil cakes and the level of their addition influenced the thermal and rheological properties of the tested systems" -  This important statement is not in the conclusion of the paper.

Response: the conclusions section was improved

Please, consider my suggestion and re-evaluate your conclusion as well.

Response: the conclusions section was improved

L.28- What was the best oil ? hazelnut or walnut ?

Response: The subject of the work was the assessment of the influence of hazelnut cake and walnut cake on the thermal and rheological properties of wheat flour. Therefore, the aim of this study was not to evaluate walnut oil or hazelnut oil.

What was the best treatment 5%, 10% 15 % ? This needs to appear in the abstract and conclusion, guys.

Response: The additional informations were added to the abstract and conclusions section.

I missed paragraphs in the text.

Response:  the paragraphs have been added in the text

  1. 36- "The first ones mentioned" - Be clearer.

Response: this sentence was rewritten

L.43- All these statements are from the same author ? Strange.... since they are about different subjects. Please, improve this part, it is really confused.

Response: the introduction to the work has been improved

L.66- Authors, it seems to me that you should first present "hazelnut and walnut", then mention both oils and their health benefits and the approach to sustainability. As it is, it's too much back and forth ... and then close with cake.

Response: The introduction has been corrected. The authors describe the health benefits of using oilseeds and the possibility of using by-products from the food industry to enrich food. Then they write about the benefits of nuts and by-products related to their production. The authors described the chemical composition and possibilities of using by-products from the oil pressing process. In our opinion, the introduction is now correct.

Would it really be by-product? What do you think of a co-product? Evaluate the real situation of the product used and the definitions.

Response: Oilseeds cakes are a co-product. As a by-product of oil pressing, oilseed cakes are produced, which are mostly used as feed for farm animals because they contain significant amounts of protein and other valuable substances [Barta et al., 2021 https://doi.org/

10.3390/foods10112766]

L.82- "Pycia and Juszczak" - These authors are 11 e not 12. 

Response: The cited researchers are co-authors of two works, marked in the literature list as 11 and 12. The citation method is correct.

L.95- A lot of important information about the preparation of the cake has not been included.

Please, add the detailed and complete formulation of all the treatments, preferably in a table to make it easier for other researchers/readers to understand. Eg : Milk ?sugar ? Whole milk ? demerara sugar ? Author details, please. The formulation must be replicable.

Response: The material section contains additional information about hazelnut and walnut cakes. Oil cakes are the remains of seeds after cold pressing of the oil. The subject of the work was the influence of partial replacement of wheat flour with hazelnut or walnut cake on the thermal and rheological properties of the obtained systems. The paper did not examine the recipe for dough with the addition of sugar, milk, etc.

What type of cake was made?  Angel cake ? Cupcake and so on...

Response: Oil cakes are the remains of seeds after cold pressing of the oil.

Baking time ? Which oven was used ? How many batches ? How many replications ?

Response: Oil cakes are the remains of seeds after cold pressing of the oil. The subject of the work was the influence of partial replacement of wheat flour with hazelnut or walnut cake on the thermal and rheological properties of the obtained systems. The paper did not examine the recipe for dough with the addition of sugar, milk, etc.

L.138- "Frequency sweep " - You used this terminology in this section and then no more. I ask you to re-evaluate all the nomenclatures used in the Material and methods and those used in the results and discussion.

Response: Subchapter titles have been corrected. In the case of the subsection "Freguency sweep" has been changed to "Determination of mechanical spectra" because the determination of the modulus G' and G" allows the assessment of viscoelastic properties, as marked in the results and discussion section.

L.154/ 155- Between or among ?

Response: Between is correct

L.317- Figure 1 and 2 need some thoughts and adjustments. The black line should also be in the caption, in my opinion. In all the results in the previous tables, you compared all the treatments. Why did you separate them in this result? I'm sorry, but I don't understand.

Response: The tables show the values of parameters illustrating the properties of all tested samples. However, the figures show the course of curves, e.g. pasting curves or mechanical spectra illustrating viscoelastic properties (G' and G” modules) separately for WFHOC and WFWOC samples. In the case of curves for all samples on one figure, the fugure would be difficult to read, because in some cases, the differences between samples are small.

L.325- Between ? Check English

Response: The manuscript was checked

L.399-  Same suggestion. In all the results in the previous tables, you compared all the treatments. Why did you separate them in this result? I'm sorry, but I don't understand.

Response: The tables show the values of parameters illustrating the properties of all tested samples. However, the figures show the course of curves, e.g. pasting curves or mechanical spectra illustrating viscoelastic properties (G' and G” modules) separately for WFHOC and WFWOC samples. In the case of curves for all samples on one figure, the fugure would be difficult to read, because in some cases, the differences between samples are small.

L.451- Follow the same reasoning, at least control, hazelnuts and walnuts with different colors.

Response: The figure 5 was improved, by the colour also.

L.424- Please, could you confirm the results 0.41a±0.01 ; 0.42b±0.00 ; 0.43ab±0.01

Response: The results were checked statistically. Now is correct.

L.484-  Same suggestion. In all the results in the previous tables, you compared all the treatments. Why did you separate them in this result?

Response: The tables show the values of parameters illustrating the properties of all tested samples. However, the figures show the course of curves, e.g. pasting curves or mechanical spectra illustrating viscoelastic properties (G' and G” modules) separately for WFHOC and WFWOC samples. In the case of curves for all samples on one figure, the fugure would be difficult to read, because in some cases, the differences between samples are small.

L.530 - I missed the limitations of the study.

Response: The conclusions of the work describe the perspective and need for continued research in the field of dough and bread fortification.

It was not good to end with the table before the conclusion.

Response: Some text has been moved below the table. It should be better now.

L.540 - " It is expected that the dough produced based on these mixtures will also have changed rheological properties" -  But this is the conclusion of your paper. It should be a solid sentence.

Response: the conclusions section has been improved

L.563 - Check Authors Guide about references. Eg  (:) DOI

Response: The literature list has been checked. Missing DOI numbers have been added.

Reviewer 4 Report

Comments and Suggestions for Authors

The manuscript under review dealt with assessing the influence of the addition of ground hazelnut oil cakes and walnut oil cakes to wheat flour on its selected thermal and rheological properties.

The materials tested (wheat flour and flour substituted with ground oil cake based on hazelnuts and walnuts) were investigated in terms of the characteristics of the gelatinization and retrogradation processes.

The manuscript is interesting and innovative because the evaluation of the rheological properties of flour or its mixtures with plant raw materials such as nut cakes is of great impact for the bakery industry. This is useful for predicting the characteristics of the dough, its behavior during kneading and the quality of the product after baking. Moreover, edible oil cakes represent a rich source of protein, fibre, minerals, vitamins and other bioactive substances, which explains why they are often used to improve the nutritional value of bakery products.

The summary is adequate and reflects the content of the work.

The introduction is comprehensive, provides sufficient background on the topic, and includes relevant references.

The objective of the study is clearly expressed.

Regarding the methodology, it is generally well detailed and appropriate. Only a few additions should be made regarding the origin and preparation of hazelnut oil cakes and walnut oil cakes.

The results are robust and strongly discussed.  The data presented in Figures 1-7 and Tables 1-5 fully support every detail discussed in the paper.

The references cited are relevant to this research topic.

 Just some concerns that should be considered:

§  In the Section 2.1. Materials, please, provide data on the origin/sourcing and preparation of hazelnut oil cake (HOC) and walnut oil cake (WOC). Was it obtained under laboratory conditions or was it supplied by an oil factory? What type of equipment was used to obtain hazelnut and walnut oil by pressing and how were the resulting by-products conditioned? Were these used immediately after obtaining or were these dried beforehand?

§  In tables and figures it is advisable to describe the abbreviations used for samples.

§  The conclusions should be improved to highlight the added value, thus strengthening the results of the manuscript.

Author Response

Reviewer 4

Comment: The manuscript under review dealt with assessing the influence of the addition of ground hazelnut oil cakes and walnut oil cakes to wheat flour on its selected thermal and rheological properties.

The materials tested (wheat flour and flour substituted with ground oil cake based on hazelnuts and walnuts) were investigated in terms of the characteristics of the gelatinization and retrogradation processes.

The manuscript is interesting and innovative because the evaluation of the rheological properties of flour or its mixtures with plant raw materials such as nut cakes is of great impact for the bakery industry. This is useful for predicting the characteristics of the dough, its behavior during kneading and the quality of the product after baking. Moreover, edible oil cakes represent a rich source of protein, fibre, minerals, vitamins and other bioactive substances, which explains why they are often used to improve the nutritional value of bakery products.

The summary is adequate and reflects the content of the work.

The introduction is comprehensive, provides sufficient background on the topic, and includes relevant references.

The objective of the study is clearly expressed.

Regarding the methodology, it is generally well detailed and appropriate. Only a few additions should be made regarding the origin and preparation of hazelnut oil cakes and walnut oil cakes.

The results are robust and strongly discussed.  The data presented in Figures 1-7 and Tables 1-5 fully support every detail discussed in the paper.

The references cited are relevant to this research topic.

Response: Thank you for your review of the work. In our opinion, the discussed topic of the use of hazelnut or walnut cakes is important and required a broader examination in terms of the rheological properties of systems containing wheat flour.

 Just some concerns that should be considered:

  • In the Section 2.1. Materials, please, provide data on the origin/sourcing and preparation of hazelnut oil cake (HOC) and walnut oil cake (WOC). Was it obtained under laboratory conditions or was it supplied by an oil factory? What type of equipment was used to obtain hazelnut and walnut oil by pressing and how were the resulting by-products conditioned? Were these used immediately after obtaining or were these dried beforehand?

Response: additional information is included in the research material section. Hazelnut or walnut cakes came from a company that professionally cold-presses oil from oilseeds. The oil was cold pressed at a temperature of 38°C. The cakes had a loose form. They were thoroughly mixed with wheat flour, and the resulting arrangements were sifted through a hand kitchen sieve. Homogeneous mixtures were stored in a cold store at 5°C until analysis.

  • In tables and figures it is advisable to describe the abbreviations used for samples.

Response: The research material section contains additional information regarding sample coding symbols. In our opinion, there is no need to repeat this information under each table and figure.

  • The conclusions should be improved to highlight the added value, thus strengthening the results of the manuscript.

Response: the conclusions section has been improved

Round 2

Reviewer 1 Report

Comments and Suggestions for Authors

Authors endorsed all of the suggestion, please add significant notation to figure 5 as well. 

Author Response

Reviewer 1

Authors endorsed all of the suggestion, please add significant notation to figure 5 as well.

Response: Figure 5 was corrected during review 1. Letters were added to indicate the presence of significant differences. The correction was performed in the track changes mode, so the deletion of the corrected figure was not visible. Now it's right.

Reviewer 2 Report

Comments and Suggestions for Authors

Authors improve the manuscript in the light of suggestions made by other esteemed reviewers.

Author Response

Reviewer 2

Authors improve the manuscript in the light of suggestions made by other esteemed reviewers.

Response: The authors thank you for this opinion.

Reviewer 3 Report

Comments and Suggestions for Authors

After another evaluation of the manuscript, I  realized some improvement in the quality of the paper. The authors have accepted almost all of my requests.

They also have improved  English, which is always useful to ask a native speaker for a final appreciation.They also correcte tables and graphs.

Some more questions :

L.28- What was the best oil ? hazelnut or walnut ?

Response: The subject of the work was the assessment of the influence of hazelnut cake and walnut cake on the thermal and rheological properties of wheat flour. Therefore, the aim of this study was not to evaluate walnut oil or hazelnut oil.

Answer: As soon as you evaluate different treatments and compare averages, I think it's important to know which one performed best.

Response: Oil cakes are the remains of seeds after cold pressing of the oil. The subject of the work was the influence of partial replacement of wheat flour with hazelnut or walnut cake on the thermal and rheological properties of the obtained systems. The paper did not examine the recipe for dough with the addition of sugar, milk, etc.

Answer: This explanation should come in the Introduction, precisely to introduce the work, I looked all over the section and couldn't find it.

Standardize all graphics "Thixotropy" are with a capital letter, in Figure 4 and 5 for example.

Please, check  all the references, I still think I've seen misquotes. Example: L.96 => reference 15 is [15] AACC. AACC Method 44-15.02 Moisture—Air-Oven Methods. AACC Approv. Methods 700 Anal. 1999.

Comments on the Quality of English Language

Minor editing of English language required

Author Response

Reviewer 3

After another evaluation of the manuscript, I  realized some improvement in the quality of the paper. The authors have accepted almost all of my requests.

They also have improved  English, which is always useful to ask a native speaker for a final appreciation.They also correcte tables and graphs.

Response: Thank you for this opinion. The manuscript has been checked and corrected in terms of language

Some more questions :

L.28- What was the best oil ? hazelnut or walnut ?

Response: The subject of the work was the assessment of the influence of hazelnut cake and walnut cake on the thermal and rheological properties of wheat flour. Therefore, the aim of this study was not to evaluate walnut oil or hazelnut oil.

Answer: As soon as you evaluate different treatments and compare averages, I think it's important to know which one performed best.

Response: The abstract and conclusions of the work provide information on the impact of replacing wheat flour with 15% of hazelnut and walnut cakes.

Response: Oil cakes are the remains of seeds after cold pressing of the oil. The subject of the work was the influence of partial replacement of wheat flour with hazelnut or walnut cake on the thermal and rheological properties of the obtained systems. The paper did not examine the recipe for dough with the addition of sugar, milk, etc.

Answer: This explanation should come in the Introduction, precisely to introduce the work, I looked all over the section and couldn't find it.

Response: The introduction describes what nut oil cakes are and how they are made. Their properties are also described and the purpose of the work is defined (lines 87 and 107).

Standardize all graphics "Thixotropy" are with a capital letter, in Figure 4 and 5 for example.

Please, check  all the references, I still think I've seen misquotes. Example: L.96 => reference 15 is [15] AACC. AACC Method 44-15.02 Moisture—Air-Oven Methods. AACC Approv. Methods 700 Anal. 1999.

Response: Axis captions on charts have been checked, standardized and corrected. Additionally, citations in the text were corrected and checked.
